# Remodeling Membrane Binding by Mono-Ubiquitylation

**DOI:** 10.3390/biom9080325

**Published:** 2019-07-31

**Authors:** Neta Tanner, Oded Kleifeld, Iftach Nachman, Gali Prag

**Affiliations:** 1School of Neurobiology, Biochemistry and Biophysics, The George S. Wise Faculty of Life Sciences, Tel Aviv University, Tel Aviv 69978, Israel; 2Faculty of Biology, Technion—Israel Institute of Technology, Haifa 32000, Israel; 3Sagol School of Neuroscience, Tel Aviv University, Tel Aviv 69978, Israel

**Keywords:** ubiquitylation, Ub receptor, phosphatidylinositol phosphate

## Abstract

Ubiquitin (Ub) receptors respond to ubiquitylation signals. They bind ubiquitylated substrates and exert their activity in situ. Intriguingly, Ub receptors themselves undergo rapid ubiquitylation and deubiquitylation. Here we asked what is the function of ubiquitylation of Ub receptors? We focused on yeast epsin, a Ub receptor that decodes the ubiquitylation signal of plasma membrane proteins into an endocytosis response. Using mass spectrometry, we identified lysine-3 as the major ubiquitylation site in the epsin plasma membrane binding domain. By projecting this ubiquitylation site onto our crystal structure, we hypothesized that this modification would compete with phosphatidylinositol-4,5-bisphosphate (PIP_2_) binding and dissociate epsin from the membrane. Using an *E. coli*-based expression of an authentic ubiquitylation apparatus, we purified ubiquitylated epsin. We demonstrated in vitro that in contrast to *apo* epsin, the ubiquitylated epsin does not bind to either immobilized PIPs or PIP_2_-enriched liposomes. To test this hypothesis in vivo, we mimicked ubiquitylation by the fusion of Ub at the ubiquitylation site. Live cell imaging demonstrated that the mimicked ubiquitylated epsin dissociates from the membrane. Our findings suggest that ubiquitylation of the Ub receptors dissociates them from their products to allow binding to a new ubiquitylated substrates, consequently promoting cyclic activity of the Ub receptors.

## 1. Introduction

Ubiquitylation regulates most cellular pathways in eukaryotes [1]. Ubiquitin (Ub) signals are written and edited by a multiplexed network of E1 (Ub-activating enzyme), E2 (Ub-conjugating enzyme), E3 (Ub-ligase), and deubiquitylating enzymes in a tightly regulated manner. Intriguingly, the same Ub molecules code many different cellular outcomes [2]. How do cells correctly interpret these Ub codes? It has become evident that hundreds of Ub receptors read and respond to Ub signals by tethering Ub binding domains (UBDs) to a “response element”. Moreover, Ub receptors possess modules that sense the cellular context (hereafter “context domain”), which allows them to function in an appropriate spatiotemporal manner. Enigmatically, Ub receptors are themselves regulated by ubiquitylation [3,4,5]. For example, in two independent studies, both Bonifacio and our own laboratory demonstrated that ubiquitylation promotes the dissociation of the Ub receptors Rabex5 and Vps9 (a human and its yeast orthologue, respectively) from their ubiquitylated cargos [6,7]. The UBDs of these Ub receptors are tethered to a GEF (guanine nucleotide exchanging factor) domain. They bind in *trans* their ubiquitylated targets on Rab5 endosomes to synchronize GEF activity with ubiquitylated cargo trafficking. Following ubiquitylation of these Ub receptors, their UBDs leave the Ub moiety on the cargo and bind the self-conjugated Ub in *cis*. It was suggested that deubiquitylation then takes place to enable the cyclic activity of these Ub receptors. Similarly, the Ub receptor Rpn10 functions as a shuttling receptor that catches the ubiquitylated targets and transports them for degradation at the proteasome. We have demonstrated that following the ubiquitylation of Rpn10 on the proteasome, its conjugated Ub clashes with Rpn9, and consequently the ubiquitylated Rpn10 dissociates from the proteasome [8]. It has been suggested that the dissociation of ubiquitylated Rpn10 followed by deubiquitylation allows the recurrent shuttling of Rpn10 with its ubiquitylated cargo to the proteasome. The above two marginally different mechanisms are both conducive to the cyclic activity of these particular Ub receptors.

Here, we asked whether the activity of epsin, a Ub receptor that functions at the plasma membrane [9], might also be regulated by ubiquitylation/deubiquitylation in order to maintain its cyclic activity. Epsin proteins decode the ubiquitylation signals of the plasma membrane proteins to promote their clathrin-dependent endocytosis [10,11,12]. In epsin, a tandem of UBDs [comprising ENTH (Epsin N-Terminal Homology domain) and two UIMs (Ub-Interacting Motif)] is linked to the response elements that recruit the endocytic machinery (Figure 1). As epsin ambushes its ubiquitylated targets at the plasma membrane, its context domain (ENTH) specifically associates with the membrane lipid phosphatidylinositol-4,5-bisphosphate (PIP_2_) [9,13]. An elegant study by McMahon and co-workers demonstrated that the amphipathic N-terminal α-helix (hereafter, H_0_) of rat epsin directly interacts with PIP_2_ and induces membrane curvature for the downstream formation of clathrin-coated pits [9]. Kozlov and co-workers calculated and demonstrated that H_0_ not only induces membrane curvature, but also contributes to membrane fission at the end of the endocytosis [14,15]. The emerging vesicles fused with early endosomes, resulting in the translocation of both the ubiquitylated cargo and epsin to early endosomes. As epsin functions at the plasma membrane, we postulated that a retrograde mechanism dissociates epsin from the early endosomes to enable cyclic functioning. Based on our studies with the Ub receptors Vps9 and Rpn10, we hypothesized that epsin ubiquitylation may also dissociate Ub–epsin from the early endosomes initiating its retrograde process. Here we describe a series of in vitro and in vivo experiments that independently tested this hypothesis.

## 2. Materials and Methods

### 2.1. Protein Expression and Purification

To purify *apo* Ent1 Rosetta2 (λDE3) BL21, *E. coli* cells were transformed with pHis_6_–Maltose Binding Protein (MBP)–Ent1 derivatives. For the purification of ubiquitylated Ent1, the same bacterial strain was co-transformed with pGEN and pCOG vectors to express the entire ubiquitylation cascade of Ent1, as described in [7]. The bacteria were grown in Terrific broth medium, supplemented with 50 µg/mL streptomycin and 34 µg/mL chloramphenicol, at 37 °C. Expression of Ent1 was induced by the addition of 0.5 mM isopropyl β-D-1-thiogalactopyranoside (IPTG), and the bacteria were grown for another ~20 h at 16 °C. Cells were harvested by centrifugation, and were re-suspended in lysozyme buffer (150 mM NaCl, 50 mM Tris–Hcl pH 7.0, and 0.1 mg/mL lysozyme), supplemented with the protease inhibitor 4-(2-aminoethyl) benzenesulfonyl fluoride hydrochloride (AEBSF). A complete lysis and DNA shearing were achieved by sonication. The soluble fraction was isolated by centrifugation. The protein was purified using nickel–nitrilotriacetic acid (Ni^2–^NTA) affinity chromatography, according to the manufacturer’s instructions. Affinity tags were clipped by rhinovirus protease during dialysis against 150 mM NaCl and 50 mM Tris–HCl pH 7.0 buffer. Subsequently, the sample was diluted to a final concentration of 35 mM NaCl supplemented with 10% glycerol, and was loaded onto an ion exchange column (SP Sepharose fast flow, GE Healthcare) pre-equilibrated with the same buffer. The column was washed, and an NaCl gradient (35–1000 mM) was applied. The protein was eluted at ~250 mM NaCl. In the case of purification of ubiquitylated-Ent1 derivatives, an additional affinity chromatography step was added using glutathione or amylose beads, depending on the specific construct, as previously described [7]. Finally, size-exclusion chromatography (with Superdex-75 16/60 HiLoad prep-grade column) was used to further increase the protein purity and to eliminate microaggregates. The protein sample was concentrated to 10 mg/mL. Then, 100 µL aliquots were flash-frozen in liquid nitrogen and preserved at –80 °C for downstream experiments.

### 2.2. In-Gel Digestion and Mass Spectrometry Analysis

Tryptic protein in-gel digestion was performed as described in [7,8,16]. Briefly, the proteins in each gel slice were reduced (10 mM dithiotreitol), modified with 40 mM iodoacetamide (at 25 °C), and trypsinized (modified trypsin (Promega)) at a 1:100 enzyme-to-substrate ratio for 18 h at 37 °C. The resulting tryptic peptides from each gel slice were resolved by reverse-phase chromatography on 0.075 × 200 mm fused silica capillaries (Aligent Technologies J&W, Santa Clara, CA, USA) packed with Reprosil reversed-phase material (Dr Maisch GmbH, Entringe, Germany). The peptides were eluted with linear 65 min gradients of 5% to 45%, and 15 min at 95% acetonitrile with 0.1% formic acid in water, at flow rates of 0.25 μL/min. Mass spectrometry (MS) was performed by an Orbitrap-XL mass spectrometer (Thermo) in a positive mode, using repetitive full MS scans followed by collision-induced dissociation of the seven most dominant ions selected from the first MS scan. Data analysis was performed using the Trans Proteomic Pipeline (TPP) version 4.6.3 [17]. TPP-processed centroid fragment peak lists in mzML format were searched against a database composed of *S. cerevisiae* yeast, *E. coli* proteins (Uniprot), and human Ub, supplemented with their corresponding decoy sequences. The database searches were performed using X! Tandem with the *k*-score plugin through TPP. Search parameters included trypsin cleavage specificity, with two missed cleavage sites; cysteine carbamidomethyl as a fixed modification; and lysine ubiquitylation (GG), methionine oxidation, and protein N-terminal acetylation as variable modifications. Peptide tolerance and MS/MS tolerance were set at 10 ppm and 0.8 Da, respectively. X! Tandem refinement included semi-style cleavages and variable lysine GG modification. Peptide and protein lists were generated following Peptide Prophet and Protein Prophet analysis, using a protein false discovery rate of 1%.

### 2.3. Structural Modeling

The crystal structure of Ent1–ENTH domain a.a. 17-150 (PDB 5LOZ), determined by Tanner, was the basis for our model [18,19]. We used Protein Homology/analogY Recognition Engine V 2.0 (Phyre^2^) to model the first 16 a.a. [20]. To obtain a complex with PI(4,5)P_2_, we superimposed the model with a rat Epsin1–ENTH domain complex with PI(4,5)P_2_ [Protein Data Bank(PDB) code 1H0A]. We manually adjusted some of the amino acids at the binding pocket with Coot [21]. Finally, the structure was idealized using energy minimization with the “Idealisation” mode of Refmac5 [22]. Figures of protein/ligand structures were prepared with PyMol [23].

### 2.4. PIP_2_-Enriched Liposome Pull-Down Assay

A mixture of 1.5 µg of apo Ent1_1-184_ and ubiquitylated-Ent1_1-184_ was incubated in 20µl of 1mM PolyPIPosomes^TM^ (Echelon Biosciences, Catalog No.: Y-P045), and diluted in 1 mL of binding buffer containing 50 mM Tris, pH 7.5, 150 mM NaCl, and 0.05% Nonidet P-40. The blend was rotated for 10 min at room temperature and centrifuged at 30,000 rpm for 15 min. The liposome pellet was resuspended in 1 ml of binding buffer, and then centrifuged. This step was repeated three times, and the bound (pellet) and supernatent samples (at a relative ratio of 10:1) were resolved by SDS-PAGE. The hemagglutinin (HA) fusion proteins were detected by western blot analysis using anti-HA antibodies (Amersham Biosciences, Little Chalfont, UK).

### 2.5. PIP Array Binding Assay

A PIP Strip (Echelon Biosciences) binding assay was performed according to the manufacturer’s instructions. Briefly, the PIP array sheet was blocked for 1 h with a 1:1 ratio of PBS:blocking buffer (LI-COR Biosciences) at room temperature. After blocking, the PIP array was incubated for 1 h with 10 mL of 1:1 ratio of PBS:blocking buffer (LI-COR Biosciences), containing 10 µg of equal amounts of Ent1_1-184_ and ubiquitylated-Ent1_1-184_. The PIP array was then washed three times and incubated with anti-His tag antibody for another hour, and then subjected to western blot analysis. The same PIP array was then incubated with anti-HA antibody for one more hour, and subjected to western blot analysis for a second detection.

### 2.6. Live Imaging of Ent1 Derivatives

*S. cerevisiae* Ent1 was PCR amplified from genomic DNA to create Ent1_4-445_, Ent1_17-445_, ENTH_4-152_, and ENTH _17-152_ constructs. Ub fusion constructs were made using a PCR sewing technique, in which Ub was amplified from pGEN25 [7]. Expression vectors harboring these constructs were made by subcloning the above sequences into the pGREG600 plasmid using a homologous recombination in yeast, and isolated vectors were sequenced [24]. The vectors were transformed into yeast strain BY4741 (MATα GFP::HIS3MX, ura3∆0, leu2∆0, his3∆1, met15∆0). The transformed cells were grown overnight at 30 °C in YPD (Yeast extract Peptone Dextrose media) with G418 selective medium (1% yeast extract, 2% peptone, 2% glucose, G418 200 μg/mL). Cultures were then washed, diluted, and grown for 3–4 h in inductive medium containing galactose (2%) and raffinose (1%) in the presence of G418 (200 μg/mL). The media was then discarded, and cells were resuspended in H_2_O. A 3–5 μL sample was put on a slide and visualized using a Nikon TiE epifluorescent microscope with an Andor Clara camera, at X100 magnification. GFP emission was imaged with 400 msec exposures.

## 3. Results

### 3.1. Yeast Ent1 Undergoes Ubiquitylation

A proteomics survey of monoubiquitylated proteins in vivo in yeast had previously demonstrated that both of the epsin proteins, Ent1 and Ent2, undergo Rsp5-dependent ubiquitylation [16]. However, probably due the limited amount of ubiquitylated-Ent1, the mass spectrometry analysis in that study could not detect the ubiquitylation sites on Ent1. In another proteomic survey in yeast, however, K103 and K148 of Ent1 and K14 of Ent2 were identified as potential ubiquitylation sites [25].

To reassess whether Ent1 undergoes mono-ubiquitylation in vivo, we transformed yeast cells with a plasmid that over-expresses RGS (Arg-Gly-Ser) –His_8_–Ub^K0^ (in which all seven lysine residues were substituted for arginine) and extracted the ubiquitylated proteins from whole-cell lysate using rapid TCA (TriChloroacetic Acid) lysis [16]. In this experiment, we used a yeast strain that expresses an Ent1–GFP (green fluorescent protein) from its native genomic locus [26]. Proteins were enriched on an NTA column, resolved by SDS-PAGE, and subjected to western blot analysis with an anti-GFP antibody (Figure 2). A band at ~80 kDa corresponding to Ent1–GFP was clearly observed in the cell extracts that both expressed and did not express the RGS–His_8_–Ub^K0^. Moreover, only in the cells that expressed the RGS-His_8_-Ub^K0^, an additional band at ~90 kDa corresponding to ubiquitylated-Ent1 was clearly observed (Figure 2). This immunoblotting analysis suggests that the Ent1 undergoes ubiquitylation in vivo, and corroborates the previous data derived from the mass spectrometry survey [16].

### 3.2. Identification of the Ent1 Ubiquitylation Sites

To uncover the biological significance of Ent1 ubiquitylation, we sought first to identify its ubiquitylation sites. Unfortunately, the amount of RGS–His_8_–Ub^K0^ ubiquitylated Ent1 derived from the yeast was again insufficient to detect the tryptic GG peptides by mass spectroscopy analysis, confirming previous results [16]. We suspect that due to the efficient deubiquitylation, the steady state amounts of modified Ent1 in vivo is insufficient for a mass spectrometry study. To circumvent this hurdle, we employed our *E. coli*-based synthetic biology approach to express the entire ubiquitylation apparatus of Ent1 [7]. In that system, we co-expressed His_6_–Ub, wheat UBA1, yeast Ubc4, yeast Rsp5, and MBP–Ent1. Using double affinity tag chromatography, we obtained milligram amounts of ubiquitylated Ent1 for downstream studies. It should be noted that we had previously demonstrated that, contrary to in vitro ubiquitylation assays that lead to spurious modification of several lysine residues of Rpn10 (for example [27,28,29]), the *E. coli*-based system faithfully recapitulates the mono-ubiquitylation on lysine 84 that is observed in vivo [7]. Similarly, several identified ubiquitylation sites of Rsp5 in our *E. coli*-based system were found to be identical with those found in in vivo studies. We performed in-gel mass spectrometry analysis of the purified *apo* and ubiquitylated-Ent1 protein bands. Interestingly, different ubiquitylation sites were observed between Ent1 constructs that contained ENTH–UIM1 and those that contained only the ENTH domain (Ent1_1-184_ and ENTH_1-152_, respectively). K3, K10, and K14, all located at the PIP_2_ binding patch, as well as K103 of the protein that also contains the UIMs (Ent1_1-184_), were identified as ubiquitylation sites, with K3 being the major site that underwent the modification (Figure 2 and Appendix A). However, in ENTH_1-152_, a construct that lacks the UIMs and consequently possesses only a single UBD, K103 and K148 were found to be the sole ubiquitylation sites. Based on our structures of *apo* yeast and zebrafish ENTH domains (PDB accession codes 5LOZ and 5LP0, respectively), and their homology to the human STAM1–VHS domain complex with Ub (PDB accession code 3LDZ), we constructed a structural model ENTH:Ub of non-covalent complexes [18,19]. The yeast ENTH:Ub complex provides a molecular explanation for the ubiquitylation sites found in the ENTH–UIMs construct, which better represent the natural structure of full-length Ent1 (Appendix A). We speculate that one of the UIMs interacts non-covalently with Ub on the Rsp5 E3-ligase, “falling back” onto the ENTH domain and promoting its ubiquitylation at the lysine-rich N-terminal sequence (H_0_). However, when the UIMs were removed, the low-affinity Ub binding patch on ENTH [18] interacted with the conjugated Ub on the Rsp5 ligase, and promoted its ubiquitylation on K103 or K148, as illustrated in Appendix A. Our structural model suggests that K103 is facing the C-terminus of Ub. K148 is highly accessible for ubiquitylation, as the terminal helix of the short construct must move to accommodate the interaction with Ub, as seen in VHS domains [30,31].

### 3.3. Structure of Ent1–ENTH Provides Insight into Plasma Membrane Binding

The structure of a rat epsin–ENTH domain in a complex with PIP_2_ (PDB accession code 1H0A) provided molecular insight into the mechanism of the lipid head recognition and membrane curvature [9]. In that study, McMahon and co-workers demonstrated that the binding of PIP_2_ to the unstructured 17 residues at the N-terminus of ENTH induces the formation of an amphipathic α-helix (H_0_) [9]. The hydrophobic face of H_0_ dissolves into the inner leaflet of the lipid bilayer and induces membrane curvature. H_0_ plays a pivotal role at both the beginning and end of endocytosis [9,15]. Positive residues at the hydrophilic face of H_0_ interact with the PIP_2_ lipid head-group (Figure 3). We determined the crystal structure of the yeast Ent1–ENTH domain that lacks the H_0_ [18,19]. Given the high homology between the yeast and the rat epsin proteins, we constructed a structural model for H_0_ (purple helix in Figure 3b). The superposition of the yeast Ent1–ENTH domain with the rat ENTH:PIP_2_ complex provided insight into the molecular model for the interaction of the yeast protein with PIP_2_ (Figure 3c,d). Binding takes place by a dozen amino acids comprising K3, R7, and K10 on αH_0_, and K14 in the proceeding loop; residues R24, the backbone atoms of T27 and S28, and residue N29 all form αH_2_, together with R62 (at αH_3_), D65, Y69 (at the proceeding loop), and H72 from αH_4_.

The structural model also sheds light on the electrostatic interactions of ENTH with PIPs. We employed the continuum solvation method APBS (Adaptive Poisson–Boltzmann Solver) with the CHARMM force field to calculate the electro-potential surface of ENTH. A positive (blue) surface located at the PIP_2_ binding pocket was clearly observed. This positive patch probably attracts the negatively-charged lipid head, and provides an additional layer of information to ensure the specificity of Ent1, in order for it to interact with the plasma membrane (Figure 3e).

### 3.4. Ubiquitylation Regulates Epsin Membrane Binding In Vitro

Based on our finding that K3, a residue located at the edge of the PIP_2_ binding site, undergoes ubiquitylation, we postulated that this modification would hinder PIP_2_ binding and consequently dissociate Ent1 from the membrane. As ubiquitylation is a covalent modification, we predicted that it would exert a strong effect in the competition for the binding site. To test this hypothesis biochemically, we conducted two independent, in vitro competition experiments for the binding of *apo* Ent1 to PIP_2_ versus ubiquitylated Ent1. Ent1 *apo* and ubiquitylated protein were purified as previously described [7,19]. First, we assessed the binding of the purified proteins to an immobilized PIP array (Figure 4). The PIP array membrane, which contains a concentration gradient of eight different phosphoinositides, was incubated with an equal quantity (1:1) mixture of *apo* Ent1_1-184_ and ubiquitylated-Ent1_1-184_, and subjected to western blot analysis. To enable differential detection of the *apo* and the ubiquitylated-Ent1 proteins, an HA tag was fused to the C-terminus of the Ent1, and an His_6_-tag to the N-terminus of Ub. The results of the western analysis with anti-His_6_ antibody were that ubiquitylated-Ent1 did not bind any of the PIPs in the array (Figure 4a). To verify that the lack of signal was not due to the failure of the anti-His antibody to detect the His_6_-tag, we spotted the ubiquitylated-Ent1 sample on the same membrane (above the PI3P gradient), and repeated the western analysis with the same anti-His antibody. Indeed, clear western blot signals were found where the ubiquitylated-Ent1 was spotted (see spots in red oval dashed line at Figure 4b). The same PIP array membrane was then incubated with an anti-HA antibody. Western blot analysis with the anti-HA clearly showed specific binding to PI(3,4)P_2_, PI(4,5)P_2_, and PI(3,4,5)P_3_, for which the latter demonstrated an apparent higher affinity (Figure 4c). These results indicate that yeast Ent1 specifically binds several phosphatidylinositol moieties, and that Ent1 ubiquitylation negatively regulates this binding.

Interestingly, our binding assays revealed that the Ent1_1-184_ presents a higher affinity to phosphatidylinositol(3,4,5)P_3_ (PIP_3_). It is worth noting that, to the best of our knowledge, *S. cerevisiae* do not possess the pathway necessary for the synthesis of PIP_3_. Our structural model suggests that the Nζ atom of K3 is located at about 5 Å from the O3 atom of the inositol, a nearly ideal distance and angle to accommodate and stabilize an additional phosphate group (not shown). In the ENTH:PIP_2_ complex, however, K3 is accessible for ubiquitylation.

To further test the above hypothesis, in the second experiment we performed a liposome pelleting assay using PIP_2_-enriched membranes. One benefit of this in vitro assay is that it better mimics the natural membrane binding. PIP_2_-enriched liposomes were incubated with a mixture of *apo* and ubiquitylated Ent1_1-184_ to enable binding. Bound and unbound proteins were then separated by sedimentation of the liposomes by ultra-centrifugation, as previously described by McMahon and co-workers [9]. Similar to the PIP-Array binding assay, we probed the pellet and the supernatant using an anti-HA antibody. We found a selective binding of only *apo* Ent1_1-184_ to the PIP_2_-enriched liposomes (Figure 4d). Specifically, the *apo* protein was clearly visible in both bound and unbound fractions (pellet and supernatant, respectively). However, the ubiquitylated Ent1 was found only in the supernatant fraction, indicating that ubiquitylation had prevented the Ent1_1-184_ from membrane binding.

### 3.5. Ubiquitylation Regulates the Plasma Membrane Association of Ent1 In Vivo

To test the hypothesis that the ubiquitylation of Ent1 regulates its membrane association in vivo, we sought to monitor the cellular localization of Ent1–GFP and constitutive ubiquitylated Ent1–GFP derivatives, using time-lapse fluorescent microscopy imaging in living cells. Deubiquitylases (DUBs) rapidly remove ubiquitylation signals, and therefore challenge such studies in eukaryotic cells. To circumvent this hurdle, we constructed a stable ubiquitylation mimicry fusion of Ent1–GFP. In these constructs, we mimicked ubiquitylation by fusing the carboxy terminus of Ub in-frame to the N-terminus of Δ3–Ent1–GFP (substituting K3; these constructs marked as Nter. Ub-fusion in Figure 5). Mimicking ubiquitylation by the fusion of Ub in-frame is sometimes valuable, especially when the authentic ubiquitylation site is located near one of the protein termini [32,33]. The high flexibility of the carboxy terminus tail of Ub probably compensates for the rigidity of the peptide bond between Ub and the N terminus of the target, compared to the iso-peptide bond formed with the ubiquitylated lysine residue. To eliminate the cleavage of Ub from the N-terminus by DUB(s) that cleave linear Ub chains, we substituted the terminal glycine with valine (Ub G76V) [34]. In the first in vivo experiment, we compared the localization of the full-length Ent1–GFP that was expressed from its natural genomic promoter [26,35] with that of the ectopic expression of Δ3–Ent1–GFP (Figure 5a,b, respectively). We found that both full-length and Δ3-Ent1 proteins are localized to the plasma membrane and to the cytosol. On the membrane, they seem to form clusters that probably represent endocytic puncta or early endosomes localization. Sometimes these proteins accumulate to the budding site, probably due to their function in recruitment of actin filament machinery (not shown).

Interestingly, the Δ3–Ent1 mutant presents a slightly more visibly stronger puncta adjacent to the plasma membrane (Figure 5b). This phenotype suggests that the elimination of K3 results in a less ubiquitylated phenotype, and consequently remains for a prolonged period on the early endosome membranes. Alternatively, it is certainly possible that the observed difference between the full-length and the Δ3–Ent1 mutant was also due to the difference in their expression levels. As stated earlier, the full-length protein is expressed from its normal chromosomal promoter, whereas the Δ3-Ent1 is expressed from a Gal promoter on a single-copy CEN (centromere) vector. Regardless of the origin of this difference, the bright phenotype of the ectopically-expressed Δ3–Ent1 is useful for assessing the possible regulation of ubiquitylation at K3 on Ent1 localization.

We next constructed several different Ent1/ENTH–GFP fusions and monitored their phenotypes (Figure 5c–j). We found that the ENTH_4-152_ domain (Figure 5c) accumulated at the plasma membrane, but did not form puncta, unlike Δ3–Ent1 (Figure 5g,j), probably due to the lack of the C-terminal tail containing the Eps15 and clathrin-binding motifs. To examine the phenotype of Ent1 that is unable to directly associate with the plasma membrane, we deleted the entire H_0_ helix by constructing ENTH_18-152_/Ent1_18-454_. Indeed, Figure 5d,h show that both ΔH_0_–ENTH and ΔH_0_–Ent1 proteins were localized only in the cytosol. These four variants provided clear visibility for membrane association versus cytosolic localization phenotypes, and therefore allowed us to assess the effect of K3 ubiquitylation in vivo. We then monitored the phenotypes of Ub fusion at the N-termini of the ENTH domain and Ent1 proteins (Ub-ENTH_4-152_ Ub-Ent1_4-454_, namely Nter. Ub-fusion), which mimic the K3 ubiquitylated form. We found that both proteins localized only to the cytosol, identical to the ΔH_0_ proteins (Figure 5e,i). This result suggests that K3 ubiquitylation hinders membrane association in vivo. We then sought to determine whether the membrane-dissociated phenotype is due to specific K3 ubiquitylation, rather than ubiquitylation per se. We therefore monitored the localization of chimeras in which Ub was fused to the C-termini (ENTH_4-152_-Ub-GFP_/_Ent1_4-454_-Ub-GFP; marked as Cter. Ub-fusion in Figure 5). We found that both fusions demonstrated membrane association and cytosolic phenotypes nearly identical to the unfused proteins (Figure 5f,j). Interestingly, it seems that the fusion of Ub at the C-terminus of Ent1_4-454_ did not significantly affect the recruitment of the endocytic machinery, as observed by the puncta phenotype. Taken together, the results of the in vitro and in vivo experiments indicate that ubiquitylation at K3 dissociates Ent1 from the membrane.

## 4. Discussion

Although the fate of ubiquitylation has been studied since the late 1970s, due to the extremely short life of ubiquitylated proteins, studies with purified ubiquitylated proteins are sparse. Our *E. coli*-based expression system for the ubiquitylation cascade allowed us to purify milligram quantities of ubiquitylated proteins, which therefore played a pivotal role in this work [7].

Based on our current data, we propose the following model (Figure 6): membrane proteins (such as transporters, receptors, or channels) usually undergo ubiquitylation/clathrin-dependent endocytosis to terminate their function. Following ubiquitylation, the Ub receptor epsin reads and decodes the ubiquitylation signal into an endocytosis response by inducing membrane curvature and recruiting other endocytic proteins, such as Eps15 and clathrin. However, during the endocytosis process epsin is trafficked together with its cargo to early endosomes. To allow epsin to function again at the plasma membrane, a retrograde mechanism is needed. Our model suggests that Rsp5-dependent ubiquitylation of epsin takes place at the early endosome membrane on lysine residues at the periphery of the PIP_2_ binding pocket. This ubiquitylation removes epsin from the membrane. We speculate that a deubiquitylation enzyme present in complex with Rsp5 then reverses the modification and allows epsin to bind the plasma membrane again.

Elegant recent structural studies by Meijers and co-workers on ENTH and ANTH domains (AP180 N-Terminal Homology) in complex with PIP_2_ suggest that these domains form oligomers on the plasma membrane [36]. They show that unlike the rat Epsin1 (1H0A), the PIP_2_ in the yeast Ent2 (5ON7) is sandwiched between two ENTH domains. Superposition of these structures revealed that the H_0_ helices form a very different orientation with respect to the rest of the PIP_2_ binding pockets. While the entire structures are well imposed with RMSD values lower than 1 Å, comparing the axis of the H_0_ helices show that they move by 53 and 59 degrees. Moreover, the PIP_2_ molecules are not superimposed. They have opposite binding orientations, and are translated by about 5 Å from each other. Most importantly, regardless of these significant structural differences of H_0_ structures, in both structures K3 is accessible for ubiquitylation (Appendix A).

It will be interesting to explore the physiological function of K3 ubiquitylation. Wendland and co-workers demonstrated that the deletion of both redundant genes *ent1* and *ent2* is lethal, but could be complemented by the expression of the ENTH domain only [37]. The authors revealed a binding site on the ENTH for GAP proteins of Cdc42, needed for cell polarity. Our live imaging movies also show the accumulation of Ent1/ENTH at a patch in the plasma membrane, marking the budding site. This finding is in line with the ENTH function identified by Wendland. The redundancy and this additional function of Ent1/2 challenge the analyses of K3 ubiquitylation and deubiquitylation in endocytosis and budding.

Considering the existence of other membrane-associated proteins and ubiquitin receptors, and in light of the current findings, we suggest that ubiquitylation-dependent regulation is not restricted to epsin, but rather may exert a broader regulatory mechanism. Indeed, Yarden and co-workers have demonstrated that mono-ubiquitylation dissociates Lst2, a membrane-associated FYVE domain (Fab-1, YOTB, Vac-1, and EEA1) containing an adaptor, from early endosomes [38]. However, Lst2 ubiquitylation takes place far from the FYVE domain, demonstrating a diversification of mechanisms for the ubiquitylation-dependent regulation of membrane binding.

## 5. Conclusions

Like enzymes, Ub receptors function in a cyclic manner. They interact with their ubiquitylated substrates and decode the signal into a unique response in a spatiotemporal manner. The current study, together with several previous ones, has found that the ubiquitylation of Ub receptors functions to dissociate them from their products, and therefore promotes their cyclic activity. This cycle of ubiquitylation and deubiquitylation therefore positively regulates the Ub receptor activity. The typical prompt association and dissociation of UBDs from Ub moieties suggests that the Ub ligase and the DUB of Ub receptors may act within a complex to accelerate their function. Indeed, many ligases were found to form complexes with deubiquitylases.

Here, we demonstrated that epsin ubiquitylation blocks the context (membrane recognition) in order to dissociate the Ub-receptor from its new cellular context, and to retrograde the Ub-receptor to its original one. Considering previous studies that have demonstrated that ubiquitylation blocks the UBD [6,7,8]; we conclude that coupled mono-ubiquitylation [5,39] on one of the three main elements (UBD, context, and response) of Ub receptors functions to dissociate the receptor from its product. We therefore suggest that ubiquitylation and deubiquitylation modulate Ub receptors, and render them in an on/off manner to promote their cyclic activity.

## Figures and Tables

**Figure 1 biomolecules-09-00325-f001:**
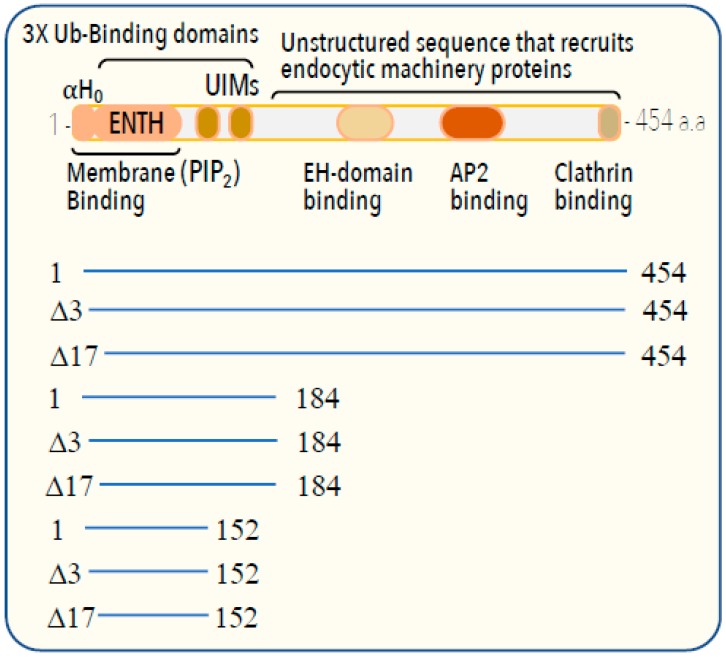
Domain architecture of epsin. The domains of yeast Ent1 and their main functions and partners are shown. Below, the main constructs examined in this study are shown.

**Figure 2 biomolecules-09-00325-f002:**
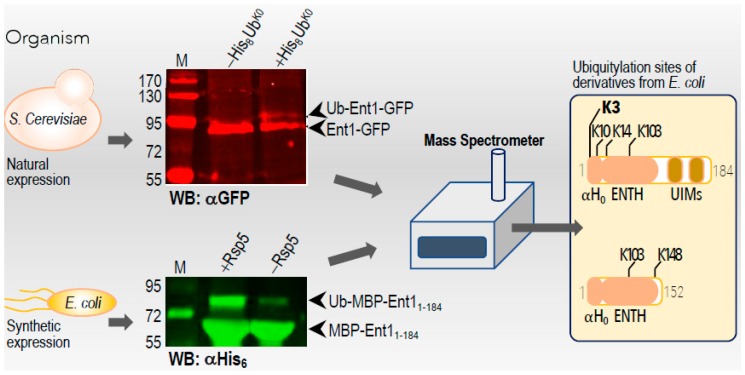
Mapping the ubiquitylation sites of Ent1. A scheme showing the approaches taken to identify the ubiquitylation sites of Ent1. *S. cerevisiae* and *E. coli* expression and purification systems described in the methods section. Purified proteins were subjected to western blot analysis with the indicated antibodies. The ubiquitylated lysine residues (Ks) identified by the mass spectrometry analysis are indicated.

**Figure 3 biomolecules-09-00325-f003:**
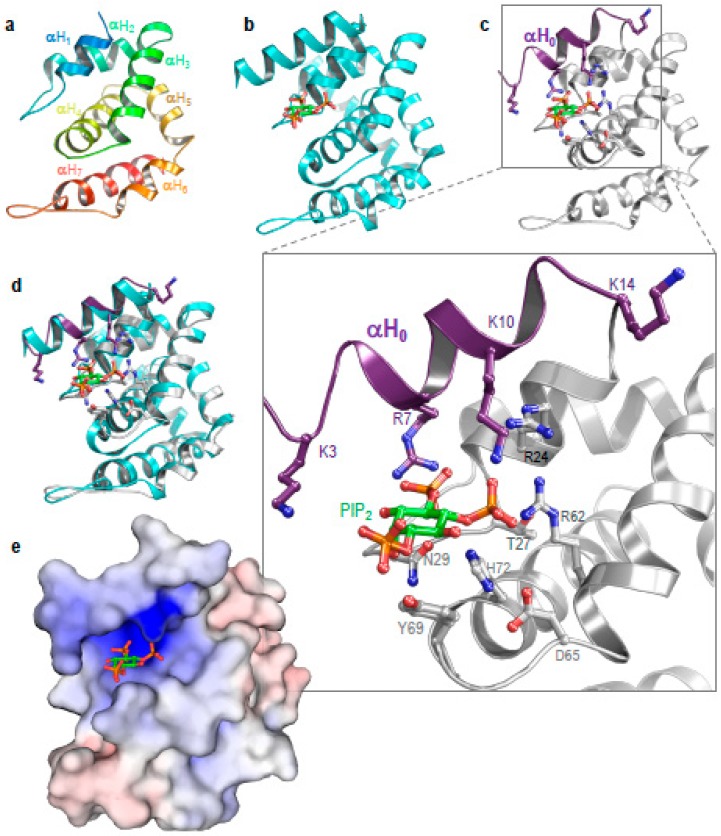
Structural insight into an ENTH domain. (**a**) The *apo* structure of Ent1-ENTH (PDB code 5LOZ). (**b**) The structure of a rat Epsin1–ENTH domain in complex with phosphatidylinositol-4,5-bisphosphate (PIP_2_; PDB code 1H0A). (**c**) A homology-based model of a yeast Ent1-ENTH domain with PIP_2_. Zooming in on the binding site shows the residues that interact with PIP_2_ and K3. (**d**) The superposition of the yeast Ent1 model with rat Epsin1. (**e**) An electro-potential surface representation of the yeast Ent1–ENTH domain. PIP_2_ is accommodated in a positively charged pocket (blue).

**Figure 4 biomolecules-09-00325-f004:**
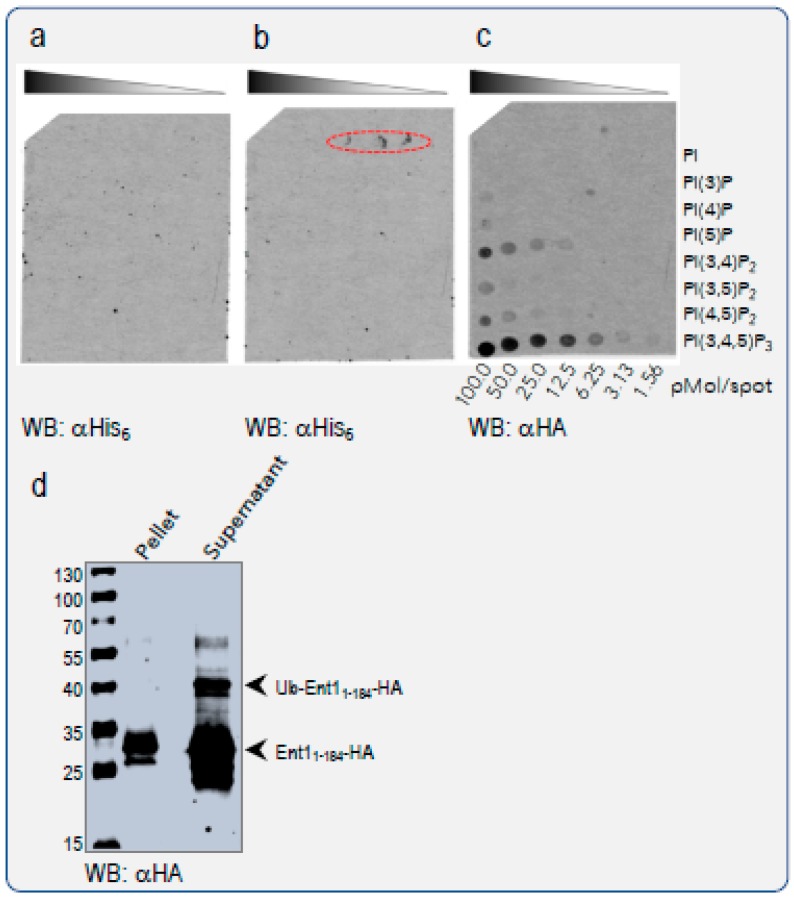
Ent1 and Ub-Ent1 PIPs binding assays in vitro. A 1:1 mix of purified *apo* and ubiquitylated Ent1_1-184_ proteins was assayed for PIPs binding. A PIP array membrane (Echelon) was incubated with the protein mix and washed, as described in the Methods section. Proteins were detected by western blot (WB): *apo* with anti-His_6_-tag, and ubiquitylated Ent1 with anti-HA antibodies, respectively. (**a**) The PIP-array membrane after a binding assay with the mix and WB against the ubiquitylated Ent1. (**b**) The antibody reaction against ubiquitylated Ent1 that was blotted on the membrane, as described in the Results section. (**c**) The reaction with the HA antibody. The binding to the indicated PIPs is clearly seen as black spots. (**d**) A PI(4,5)P_2_ enriched liposome (Echelon) binding assay. A protein mix contained *apo* and ubiquitylated Ent1_1-184_ was used for a pull-down assay with PI(4,5)P_2_ enriched liposomes. Following incubation, the liposomes were pelleted by centrifuge. Bound (Pellet) and unbound (Supernatant) proteins were then separated by SDS-PAGE, and detected by WB with anti-HA antibodies.

**Figure 5 biomolecules-09-00325-f005:**
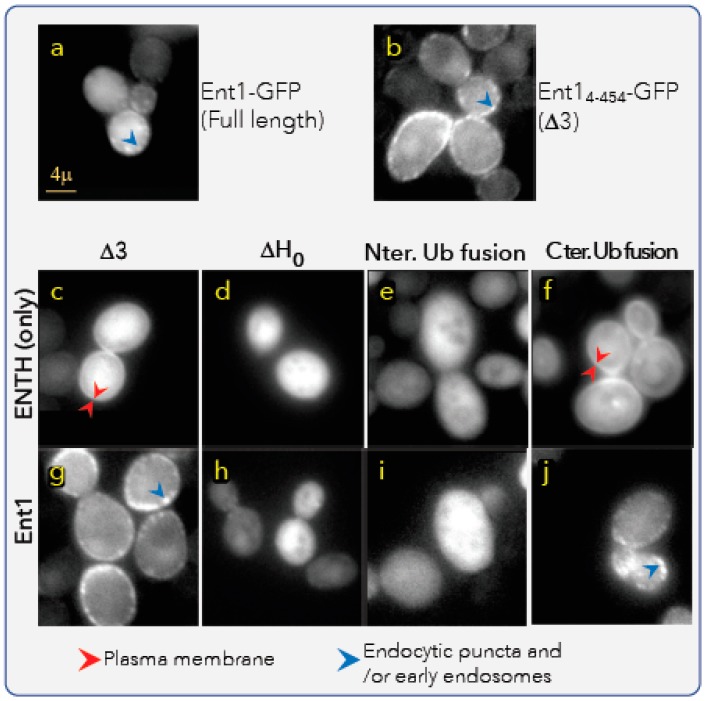
Ent1 and Ub–Ent1 membrane binding assays in vivo. Fluorescent microscopy imaging of living yeast cells expressing GFP fusion of the indicated Ent1 derivatives are shown. (**a**) A full-length Ent1 with C-terminus in-frame fusion GFP, expressed from the native genomic promoter. (**b**) An Ent1_4-454_ (Δ3) with C-terminus in-frame fusion GFP, expressed from a Gal promoter on a single copy plasmid. (**c**–**j**) The fluorescent readouts from indicated derivatives. ΔH_0_ is the construct that lacks the first 17 residues. In the Ub-fusion constructs, Ub was fused in-frame instead of K3 (i.e., to residue Q4). In the fusion-Ub constructs, Ub is fused in-frame at the C-terminus of the Ent1 derivative (between Ent1 and GFP). Red arrows mark probably endocytic or early endosome clusters of Ent1–GFP; yellow arrows mark the accumulation of Ent1-GFP fusions at the plasma membrane.

**Figure 6 biomolecules-09-00325-f006:**
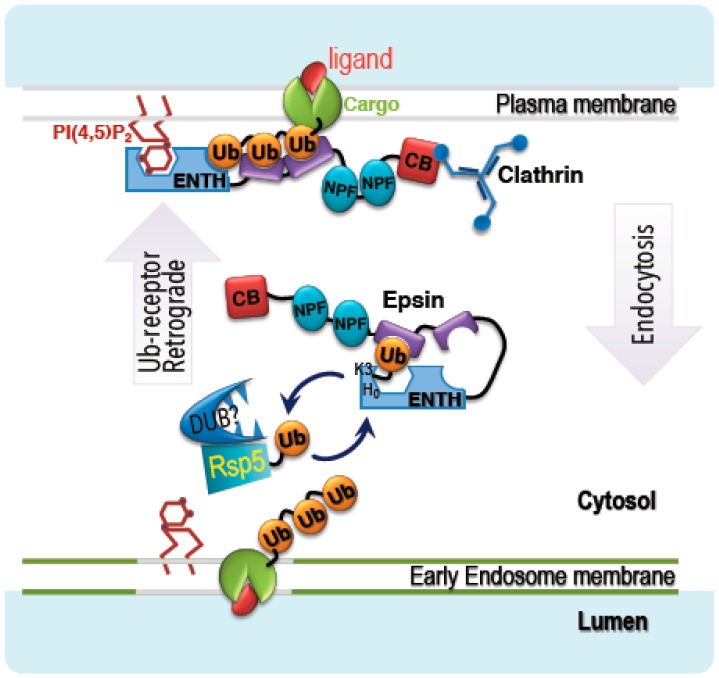
A model for the function of Ent1 ubiquitylation to initiate a retrograde process. Ent1 binds to the plasma membrane via the interaction of the ENTH domain with PIP_2_. The three UBDs of Ent1 bind to ubiquitylated cargo, and the clathrin binding box (CB) recruits clathrin proteins to promote endocytosis. For simplicity, other endocytic components (such as Eps15 that binds NPF modules) are not shown. Ent1 ubiquitylation takes place on early endosomes, probably by Rsp5, a process that dissociates Ent1 from the membrane. An uncharacterized deubiquitylase (DUB) that probably forms a complex with Rsp5 then removes Ub from ubiquitylated Ent1 to enable the cyclic function of Ent1.

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
