# Peer review of "Remodeling Membrane Binding by Mono-Ubiquitylation"

_biomolecules, 2019, doi:10.3390/biom9080325_

Round 1

Reviewer 1 Report

In this paper, the authors have described the potential function of ubiquitination of yeast Epsin, a ubiquitin receptor, and identified the main ubiquitination sites of this protein. They show that ubiquitination (on Lys3) appears to prevent the binding of Epsin onto membranes. The authors thus suggest that ubiquitin modification of Ub receptors is a mechanism to regulate their binding onto their substrates and to regulate their activity.

This manuscript is to the most part well done, but I have a few remarks:

Major:

Figure 4d: The authors claim that the data from their liposome binding assay shows that  since the ubiquitin-modified Ent1 (Epsin) fragment is not present in the pellet fraction, this thus indicates that the ubiquitinated form does not bind to the membranes. However, there is a huge difference in the intensities of the non-ubiquitinated (lower) vs. ubiquitinated (upper band) forms in the supernatant fraction, whereas the signal for the non-ubiquitinated form in the pellet is very much less. If the same difference in the intensities lower vs. upper band were present in the pellet, I doubt if the signal for the upper band would be visible here (just because it is too faint)! The authors should try to provide a more balanced blot by loading less of the supernatant fraction and aim at an exposure where a) the intensities of the lower bands in sup and pellet are about the same and b) the upper band in the supernatant is clearly visible.  

Figure 5: The images are pretty messy and it is impossible to see any details! Please make the IF images larger by omitting the frame and using smaller font on the left. I think it is also not necessary to show this many cells per image.

At several points, the manuscript suffers from bad English, here just some examples:

R.40: “Followed ubiquitination…” (should be “following”)

R. 42: “to enabling the cyclic activity” (should be “to enable”

Minor:

Please use correct definition for GEF: guanine nucleotide exchange factor; not exchanging factor!    

The authors state in the materials and methods that the liposomes were pelleted with 15000 rpm. On the other hand, they state in Results and Fig. 4 Legend that this was ultracentrifugation, although any lab centrifuge can do 15000 rpm. Please check if the rpm is correct!  

Author Response

Comments and Suggestions for Authors

In this paper, the authors have described the potential function of ubiquitination of yeast Epsin, a ubiquitin receptor, and identified the main ubiquitination sites of this protein. They show that ubiquitination (on Lys3) appears to prevent the binding of Epsin onto membranes. The authors thus suggest that ubiquitin modification of Ub receptors is a mechanism to regulate their binding onto their substrates and to regulate their activity.

This manuscript is to the most part well done, but I have a few remarks:

Major:

Figure 4d: The authors claim that the data from their liposome binding assay shows that  since the ubiquitin-modified Ent1 (Epsin) fragment is not present in the pellet fraction, this thus indicates that the ubiquitinated form does not bind to the membranes. However, there is a huge difference in the intensities of the non-ubiquitinated (lower) vs. ubiquitinated (upper band) forms in the supernatant fraction, whereas the signal for the non-ubiquitinated form in the pellet is very much less. If the same difference in the intensities lower vs. upper band were present in the pellet, I doubt if the signal for the upper band would be visible here (just because it is too faint)! The authors should try to provide a more balanced blot by loading less of the supernatant fraction and aim at an exposure where a) the intensities of the lower bands in sup and pellet are about the same and b) the upper band in the supernatant is clearly visible.

The experiment was carried out with a combination mix  of apo- and ubiquitylated-Epsin proteins incubated with PI(4,5)P2 enriched liposomes. Therefore, binding may take place for both protein forms. The relatively low amount of non-ubiquitylated (apo) Ent1 in the pellet is probably a result of the low capacity of the liposomes’ binding sites. We observed that after pelleting the liposomes only the apo-Epsin was found in the fraction. We agree that there was less modified-Ent1; however,  there was nevertheless more than enough to detect binding of the modified protein if this had occurred.  The finding suggests that the ubiquitylated-Epsin does not bind the liposomes.

To facilitate the readers’ understanding we have now clarified this important point in the text (Lines 261-271 and 490-493).

Figure 5: The images are pretty messy and it is impossible to see any details! Please make the IF images larger by omitting the frame and using smaller font on the left. I think it is also not necessary to show this many cells per image.

We thanks the reviewer for this very helpful and constructive comment. We have revisited the movies and selected better images. We followed the reviewer’s comments and have made the images larger (by cropping smaller fields) and increased the zoom. We feel that the figure indeed now better conveys the accumulated data and significantly facilitates the delivery of the story.    

At several points, the manuscript suffers from bad English, here just some examples:

R.40: “Followed ubiquitination…” (should be “following”)

R. 42: “to enabling the cyclic activity” (should be “to enable”

The English has been revised and corrected

Minor:

Please use correct definition for GEF: guanine nucleotide exchange factor; not exchanging factor!

This has been corrected as suggested.

The authors state in the materials and methods that the liposomes were pelleted with 15000 rpm. On the other hand, they state in Results and Fig. 4 Legend that this was ultracentrifugation, although any lab centrifuge can do 15000 rpm. Please check if the rpm is correct!  

We thank  the reviewer for this comment as there was indeed an error in the methods; we were indeed using a mini-ultra-centrifuge at 30K rpm. We have corrected the text accordingly.

Reviewer 2 Report

This is a very interesting subject to those who study ubiquitin signalling. The manuscript is relevant to many examples of ubiquitin receptors that contain UBDs/UIMs that read ubiquitin signals and are themselves ubiquitylated. This could be a unifying mechanism for the regulation of numerous ubiquitin signalling readers. The manuscript is focused on signalling events at the plasma membrane and endocytosis specifically, but it could be relevant to other ubiquitin signalling pathways. 

The manuscript is well written and would appeal to the readership of Biomolecules as well as many others interested in ubiquitin signalling. Before publication, the authors should consider addressing the following points to improve the clarity of their findings.

-- Major point

The authors mention in line 204 that they constructed a structural model of ENTH:Ub non-covalent complexes but do not refer to this in any of their figures. This section was very difficult to follow without a figure showing the proposed models describing the molecular roles of the ubiquitylation sites. Since the structures of ENTH and STAM-VHS domains are published, a cartoon illustration may be more appropriate. The predicted movement of UIMs in the "falling back" position needs to be illustrated too, and the cartoon illustration should indicate the position of K103 and K148 to show their accessibility. It is really hard to grasp the level of detail for this model without illustration. 

-- Minor point

Line 165 -- Can the authors specifically state "mono-ubiquitylated" proteins instead of "ubiquitylated". This would avoid confusion since His8-UbK0 should not form poly-Ub chains.   

Author Response

Comments and Suggestions for Authors

This is a very interesting subject to those who study ubiquitin signalling. The manuscript is relevant to many examples of ubiquitin receptors that contain UBDs/UIMs that read ubiquitin signals and are themselves ubiquitylated. This could be a unifying mechanism for the regulation of numerous ubiquitin signalling readers. The manuscript is focused on signalling events at the plasma membrane and endocytosis specifically, but it could be relevant to other ubiquitin signalling pathways. 

The manuscript is well written and would appeal to the readership of Biomolecules as well as many others interested in ubiquitin signalling. Before publication, the authors should consider addressing the following points to improve the clarity of their findings.

-- Major point

The authors mention in line 204 that they constructed a structural model of ENTH:Ub non-covalent complexes but do not refer to this in any of their figures. This section was very difficult to follow without a figure showing the proposed models describing the molecular roles of the ubiquitylation sites. Since the structures of ENTH and STAM-VHS domains are published, a cartoon illustration may be more appropriate. The predicted movement of UIMs in the "falling back" position needs to be illustrated too, and the cartoon illustration should indicate the position of K103 and K148 to show their accessibility. It is really hard to grasp the level of detail for this model without illustration. 

This is a highly important comment.  As the model of ENTH:Ub complex and its mutational analyses has been published (Levin-Kravets et al. Nature Methods 2016), we considered   reproducing the model to be redundant. However, the reviewer emphasizes the need to demonstrate the accessibility of the lysin residues that undergo ubiquitylation. We have added a new figure, supplementary figure 2, showing the model ENTH:Ub complex, key residues at the interface, and the relevant lysine residues. We do not show the UIMs as we find this to be highly speculative. Accordingly, we have changed the text from “We suggest” to “We speculate”    

-- Minor point

Line 165 -- Can the authors specifically state "mono-ubiquitylated" proteins instead of "ubiquitylated". This would avoid confusion since His8-UbK0 should not form poly-Ub chains.   

We thank the reviewer for this comment. Indeed we were looking for mono-ubiquitylation. We have corrected the text as suggested.

Reviewer 3 Report

In this paper, the authors identify Lys3 as a ubiquitination site of yeastEnt1 and show biochemically that Lys3 ubiquitination interferes with PIP2 binding of the ENTH domain. Although the authors do not show the evidence that Lys3 ubiquitination occurs in yeast, ectopic expression of Lys3-Ub-mimicking Ent1 proteins suggests that this ubiquitination results in dissociation of Ent1 from the membranes.

This study provides a new insight into the regulation of Ub receptors. To strengthen the paper, the authors should address the following issues.

The authors show that the role of Lys3 ubiquitination in membrane dissociation of Ent1 using the Ub-fusion proteins. Did the authors investigate whether the Ub fusion and Lys3 deletion affect endocytosis?

Figure 4d: There is no negative control data for Ent1-184-HA. The authors
should perform the same experiment using other PIP liposomes, such as PI(4)P.

Figure 5, c-j: It is difficult to distinguish between plasma membrane localization and cytosolic localization. Also, endosomal localization of Ent1-Ub is not clear (panel j). The authors should clearly show the localization of the GFP-fusion proteins (i.e. confocal microscopy or costaining with maker proteins/dyes).

Figure 4c: Are the labels of PIPs out of position?

Figure 5, b and g: These images are identical.

Lines 300 and 337: “Ub-fusion” is missing in Figure 6.

Author Response

Comments and Suggestions for Authors

In this paper, the authors identify Lys3 as a ubiquitination site of yeast Ent1 and show biochemically that Lys3 ubiquitination interferes with PIPbinding of the ENTH domain. Although the authors do not show the evidence that Lys3 ubiquitination occurs in yeast, ectopic expression of Lys3-Ub-mimicking Ent1 proteins suggests that this ubiquitination results in dissociation of Ent1 from the membranes.

This study provides a new insight into the regulation of Ub receptors. To strengthen the paper, the authors should address the following issues.

The authors show that the role of Lys3 ubiquitination in membrane dissociation of Ent1 using the Ub-fusion proteins. Did the authors investigate whether the Ub fusion and Lys3 deletion affect endocytosis?

We thank the reviewer for this valuable comment/question. As Ent1 is redundant with Ent2, none of the genes are essential for endocytosis and viability. Wendland and co-workers demonstrated that deleting both genes (ent1/ent2) is lethal. Moreover, she also demonstrated that the ENTH domain is both necessary and sufficient for viability of ent1/ent2 deleted cells. Wendland revealed an essential patch on the ENTH domain that binds Cdc42 GAP proteins and is needed for cell polarity. This evidence challenges the analysis of endocytosis-dependent ubiquitylation/deubiquitylation at K3. We therefore consider this to be beyond the scope of the present manuscript. We have added a paragraph in the Discussion (starts on line 348) that describes this issue in respect to our findings.

Figure 4d: There is no negative control data for Ent1-184-HA. The authors
should perform the same experiment using other PIP liposomes, such as PI(4)P.

Our PIP-array binding assay demonstrated that Ent1 does not bind PI(4)P;  hence we do not expect  to see any differences between apo and ubiquitylated Ent1. Nevertheless, it would be nice to have such a negative control.  Unfortunately, due to lack of manpower and the limited given time for the corrections of the manuscript we cannot repeat this experiment with this control. Taking together the entire in-vitro and in-vivo data, we feel that the integrity of the model remains regardless of this missing control

Figure 5, c-j: It is difficult to distinguish between plasma membrane localization and cytosolic localization. Also, endosomal localization of Ent1-Ub is not clear (panel j). The authors should clearly show the localization of the GFP-fusion proteins (i.e. confocal microscopy or costaining with maker proteins/dyes).

We greatly appreciate this as a highly important comment, which was also commented by ref #1. Based on the suggestions of both referees we have made a new panel of Figure 5 to address these valuable comments. In the revised figure the fields are smaller and display a lower number of cells, hence enabling higher magnification. The membranes and the puncta are now significantly more visible. 

Figure 4c: Are the labels of PIPs out of position?

The labels are aligned with the spots

Figure 5, b and g: These images are identical.

I thank the reviewer for their attention to this problem. In the new set of images we have corrected it.

Lines 300 and 337: “Ub-fusion” is missing in Figure 6.

We thank the reviewer for this valuable comment. Ub-fusion at the N-terminus mimics the normal ubiquitylation at K3. In our model (Figure 6) the ubiquitylated form of Epsin is shown in the middle. However, K3 is indeed not  indicated. In the revised figure we have added K3. 

Round 2

Reviewer 1 Report

The authors did not yet correct the definition of GEF although they state in their rebuttal to have done this. This should be done upon correction of the proofs!

Reviewer 3 Report

Although the authors did not repeat the PIP-liposome-binding assay with negative control, the manuscript is improved.